# [Reproducibility Report] Explainable Deep One-Class Classification

## Reproducibility Summary

**Scope of Reproducibility**

Liznerski et al. [23] proposed Fully Convolutional Data Description (FCDD), an explainable version of the Hypersphere Classifier (HSC) to directly address image anomaly detection (AD) and pixel-wise AD without any post-hoc explainer methods. The authors claim that FCDD achieves results comparable with the state-of-the-art in sample-wise AD on Fashion-MNIST and CIFAR-10 and exceeds the state-of-the-art on the pixel-wise task on MVTec-AD. They also give evidence to show a clear improvement by using few (1 up to 8) real anomalous images in MVTec-AD for supervision at the pixel level. Finally, a qualitative study with horse images on PASCAL-VOC shows that FCDD can intrinsically reveal spurious model decisions by providing built-in anomaly score heatmaps.

**Methodology**

We have reproduced the quantitative results in the main text of [23] except for the performance on ImageNet: sample-wise AD on Fashion-MNIST and CIFAR-10, and pixel-wise AD on MVTec-AD. We used the author's code with GPUs NVIDIA TITAN X and NVIDIA TITAN Xp. A more detailed look into FCDD's performance variability is presented, and a Critical Difference (CD) diagram is proposed as a more appropriate tool to compare methods over the datasets in MVTec-AD. Finally, we study the generalization power of the unsupervised FCDD during training.

**Results**

All per-class performances (in terms of Area Under the ROC Curve (ROC-AUC) [31]) announced in the paper were replicated with absolute difference of at most 2% and below 1% on average, confirming the paper's claims. We report the experiments' GPU and CPU memory requirements and their average training time. Our analyses beyond the paper's scope show that claiming to "exceed the state-of-the-art" should be considered with care, and evidence is given to argue that the pixel-wise unsupervised FCDD could narrow the gap with its semi-supervised version.

**What was easy**

The paper was clear and explicitly gave many training and hyperparameters details, which were conveniently set as default in the author's scripts. Their code was well organized and easy to interact with.

**What was difficult**

Using ImageNet proved to be challenging due to its size and need to manually set it up; we could not complete the experiments on this dataset.

**Communication with original authors**

We reached the main author by e-mail to ask for help with ImageNet and discuss a few practical details. He promptly replied with useful information.

## 1  Introduction

Liznerski et al. [23] proposed a deep learning based AD method capable of doing pixel-wise AD (also known as "anomaly segmentation") by directly generating anomaly score maps with a loss function based on the Hypersphere Classifier (HSC) [29], a successor of Deep Support Vector Data Description (DSVDD) [28], using a fully-convolutional neural network – hence the name Fully Convolutional Data Description (FCDD).

By only using convolutions, down-samplings, and batch normalization (no attention mechanism, nor fully connected layers), an image of dimensions $C \times H \times W$ (respectively, the number of channels, the height, and the width) is transformed into a latent representation $C' \times U \times V$, where $U < H$ and $V < W$. This low-resolution representation is a $U \times V$ grid of $C'$-dimensional vectors, from which the pseudo-Huber loss function yields a $U \times V$ heatmap of anomaly scores.

Each of these $C'$-dimensional vectors contains information from a corresponding receptive field within the full resolution ($H \times W$) image. Evidence [24] suggests that the effective influence of the input pixels decreases Gaussian-ly as their position is further away from the center of the receptive field. FCDD uses this principle to up-sample the obtained heatmap back to the original resolution ($H \times W$), therefore directly obtaining a visual, explainable anomaly score map.

Finally, FCDD is also adapted to perform anomaly detection at the sample (image) level by taking the average score on the low resolution anomaly heatmap.

**Vocabulary: Sample v.s. Pixel-wise Anomaly Detection**   The authors refer to anomaly detection (AD) at the image level (e.g. given an unseen image, a model trained on horse images should infer if there is a horse present in it or, otherwise, the image is anomalous) simply as "detection", while anomaly segmentation/localization (i.e. finding regions, sets of pixels where there exists anomalous characteristics) is referred as "pixel-wise AD". Analogously, for the sake of clarity, we refer to the former as "sample-wise AD". Both setups are further explained in Section 3.3.

## 2  Scope of reproducibility

We aimed to reproduce the results announced in [23] to verify the effectiveness of the proposed method both in sample-wise and pixel-wise anomaly detection. Specifically, we tested the following claims from the original paper:

1. **Claim 1**: FCDD is comparable with state-of-the-art methods in terms of ROC-AUC in sample-wise anomaly detection on standard benchmarks (namely, Fashion-MNIST, CIFAR-10, and ImageNet);

2. **Claim 2**: FCDD exceeds the state-of-the-art on MVTec-AD in anomaly segmentation in the unsupervised setting in terms of pixel-wise ROC-AUC;

3. **Claim 3**: FCDD can incorporate real anomalies, and including only a few annotated images ($\approx 5$) containing real, segmented anomalies, the performance consistently improves;

4. **Claim 4**: FCDD can reveal spurious model decisions without any extra explanation method on top of it.

The experiments on supporting **Claim 1** on Fashion-MNIST and CIFAR-10 have been replicated, as well as all the tests on MVTec-AD, supporting **Claims 2 and 3**, and the qualitative analysis on PASCAL-VOC, supporting **Claim 4**. We provide details about computational requirements (CPU memory, GPU memory, and training time) necessary to run these experiments.

**Beyond the paper**   Other analyses are proposed on the results obtained from the experiments corresponding to **Claims 2 and 3**, which further confirm **Claim 3** but show that **Claim 2** should be taken with consideration. We also investigate the evolution of the test performance during the optimization in MVTec-AD's unsupervised setting (see Section 3.3), revealing opportunity for improvement that could narrow down the gap with the semi-supervised setting.

## 3  Methodology

We used the author's code (PyTorch 1.9.1 and Torchvision 0.10.1), publicly available on GitHub [4], to reproduce the quantitative experiments presented in the main text. It required no external documentation, and the whole reproduction took roughly one month-person of work.

## 3.1 Datasets

The proposed method was originally tested [23] on Fashion-MNIST [32], CIFAR-10 [19], ImageNet1k [13], MVTec-AD [9], and PASCAL VOC [14]. Besides, EMNIST [11], CIFAR-100 [19], and ImageNet21k[1] (version "fall 2011") were used as Outlier Exposure (OE) [16] datasets. All the datasets except for ImageNet were publicly available and automatically downloaded.

**ImageNet**  We requested access and download ImageNet1k (version "ILSVRC 2012") from its official website [5]. ImageNet21k (a.k.a. ImageNet22k) was downloaded from `academictorrents.com` [1] because the version used in the original paper was not available in the official website anymore.

## 3.2 Models

We used the same neural networks as the original paper, which depend on the dataset:

- **Fashion-MNIST**: three convolutional layers separated by two max-pool layers, where the first convolution is followed by a batch normalization and a leaky ReLU;
- **CIFAR-10**: two convolutions preceded by three blocks, each, composed of a convolution, a batch normalization, a leaky ReLU, and a max-pool layer;
- **MVTec-AD and PASCAL-VOC (Clever Hans)**: the first 10 (frozen) layers from VGG11 pre-trained on ImageNet followed by two convolutional layers.

## 3.3 Experimental setup

The paper presents two quantitative experiments: sample-wise (section "4.1 Standard Anomaly Detection Benchmarks" in [23]) and pixel-wise AD (section "4.2 Explaining Defects in Manufacturing" in [23]), as well as a qualitative experiment.

We followed the same experimental procedure used in [23]: each experiment – i.e. given a dataset, its OE when applicable, a normal class, and all hyperparameters – was repeated five times, and the reported values are the average over them unless stated otherwise (e.g. Figure 1).

**Sample-wise**  Standard one-vs-rest setup, where one class of the given database is chosen as normal and all the others are used as anomalous. Each image has a binary ground truth signal – logically derived from its label – and the model assigns an anomaly score to it (therefore "sample-wise"). The metric used is the ROC-AUC on the test split, and all the classes are evaluated as normal. The datasets used in this experiment and their respective OE dataset is summarized in Table 1, and its results support **Claim 1**.

Table 1: Sample-wise experiments: tested datasets and their respective OE sources. From the dataset in the column "one-vs-rest", one class is used as normal at training and test time while all others are considered anomalies at test time only. The column Outlier Exposure (OE) is the dataset used as a source of anomalies at training time. "Experiment reference" will further be used to reference these configurations.

| One-vs-rest dataset | OE dataset | Experiment reference |
|---|---|---|
| Fashion-MNIST | EMNIST | F-MNIST (OE-EMNIST) |
| | CIFAR-100 | F-MNIST (OE-CIFAR-100) |
| CIFAR-10 | CIFAR-100 | CIFAR-10 |
| ImageNet1k | ImageNet21k | ImageNet |

**Pixel-wise**  Anomalies are defined at the pixel level (binary segmentation mask where "1" means "anomalous" and "0" means "normal") and an image is considered anomalous if it *contains* anomalous pixels although normal pixels are also present in the image. In each experiment a single class in MVTec-AD is fixed, its normal images are used both for training and test, and anomalous ones are used for test. As for the anomalous samples at training time, two settings were tested:

---

[1]Also known as "ImageNet22k" or "full ImageNet".

Table 2: Memory requirements and training time using NVIDIA GPUs TITAN X and TITAN Xp (one at a time, indistinctly).

| Experiment | CPU memory (Gb) | GPU memory (Gb) | Training duration |
|---|---|---|---|
| F-MNIST (OE-CIFAR-100) | 2 | 1.3 | 12 min |
| CIFAR-10 | 3 | 1.9 | 34 min |
| MVTec-AD unsupervised | 38 | 5.5 | 1h 13 min |
| MVTec-AD semi-supervised | 33 | 5.5 | 41 min |
| PASCAL VOC (Clever Hans) | 5 | 11.8 | 21 min |

- **Unsupervised**: synthetic random anomalies are generated using a "confetti noise" (colored blobs added to the image);

- **Semi-supervised**: one image per anomaly group (1 up to 8 types depending on the class) is removed from the test set and used for training.

Neither of these settings require an OE dataset because the anomalous samples are either synthetic or real on images of the nominal class. The performance metric is the ROC-AUC of the anomaly scores at the pixel level. MVTec-AD is the only dataset used in this case, and the results of these experiment support **Claims 2 and 3**.

**Clever Hans (PASCAL VOC)** About one fifth of the images in the class "horse" in PASCAL VOC [14] contain a watermark [20], which may cause models to learn spurious features. This is known as the "Clever Hans" effect as a reference to Hans, a horse claimed to be capable of performing arithmetic operations while it, in fact, read his master's reactions [26] – analogously, a model making decisions based on the watermarks would be "cheating" the real problem. In this experiment, a model is trained using all the classes in PASCAL VOC as normal, only the class "horse" as anomalous (a swapped one-vs-rest setting), and ImageNet1k as OE dataset. The goal is to qualitatively observe if one class classifiers are also vulnerable to the Clever Hans effect and show that FCDD transparently reveals such weaknesses as it intrinsically provides explanations (score heatmaps). This experiment has no quantitative metric but it supports **Claim 4**.

### 3.4 Hyperparameters

Running the author's code with the default parameters, as described in the original paper, did not require any hyperparameter tuning to achieve the reported results (differences detailed in Section 4) and confirm the authors' claims. We underline that the results on MVTec-AD were obtained using the same hyperparameters on both settings, unsupervised and semi-supervised.

### 3.5 Computational requirements

We used the NVIDIA GPUs "TITAN X" [6] and "TITAN Xp" [7] to run our experiments. The two GPUs were used indistinctly as they have similar characteristics, and only one GPU was used at a time. The GPU and CPU memory requirements and the average training duration of our experiments are listed in the Table 2 below.

CPU memory was recorded with an in-house python script using the library `psutil` [3] at 1Hz. GPU memory was recorded using `gpustat` [2] at 1Hz. Both memory values are the maximum recorded during the experiments, including training and inference time. The training duration is an average over all the experiments.

On F-MNIST (OE-CIFAR-100) and CIFAR-10, the range of duration did not vary more than two minutes, on MVTec-AD unsupervised it ranged from 15 minutes up to one hour, and on MVTec-AD semi-supervised it ranged from 22 minutes up to one hour and 56 minutes depending on the class.

### 3.6 Beyond the paper

We propose a more detailed visualization of the distribution of performances (due to random effects, all hyperparameters held constant) of the two settings (unsupervised and semi-supervised) evaluated on MVTec-AD, and a critical difference diagram as alternative evaluation of performance across several datasets (the individual classes in MVTec-AD).

The network architectures used for the experiments on MVTec-AD were pre-trained on ImageNet and most of the weights are kept frozen, therefore raising the question of how much of FCDD's performance is due to the pre-training. We took snapshots of the unsupervised model's weights in order to visualize the evolution of the performance on the test set during training.

## 4 Results

### 4.1 Reproducing the original paper

We reproduced the unsupervised and semi-supervised settings for MVTec-AD, and all experiments on Table 1 except for ImageNet – due to resource limitations, this experiment could not be completed in time.

The results of F-MNIST (OE-EMNIST) were not detailed in the original paper but it is claimed to have a class-mean ROC-AUC of $\sim 3\%$ below F-MNIST (OE-CIFAR-100), and we observed a difference of 2.7%.

We summarize the differences between our results and those from the original paper [23] in Table 3. The error margins presented are in absolute differences and refer to the ROC-AUC (which is expressed in %) from each individual class's experiment (recall: the mean over five iterations).

Table 3: Differences between the original paper results and ours. All the values are in absolute difference of ROC-AUC, expressed in % (it is **not** a relative error). The columns in "Difference per class" show statistics of the absolute difference of each individual class performance, while the column "Mean ROC-AUC diff." corresponds to the difference measured after the mean is taken over all the classes.

| Experiment | ROC-AUC type | N. classes | Diff. per class | | Mean ROC-AUC diff. |
|---|---|---|---|---|---|
| | | | Max | Mean | |
| F-MNIST (OE-CIFAR-100) | sample-wise | 10 | 1% | 0.6% | 0.01% |
| CIFAR-10 | sample-wise | 10 | 0.5% | 0.3% | 0.4% |
| MVTec-AD unsupervised | pixel-wise | 15 | 2% | 0.6% | 0.2% |
| MVTec-AD semi-supervised | pixel-wise | 15 | 2% | 0.7% | 0.4% |

**Clever Hans (PASCAL VOC)**   The experiment on PASCAL VOC ("Clever Hans Effect") has been manually verified, and similar (flawed) explanations on horse images have been observed. Two examples are shown in Figure 5 in Appendix A.

### 4.2 Beyond the paper

Figure 1 further details the performance comparison between the unsupervised and semi-supervised settings on MVTec-AD on each class.

Figure 2 compares the methods in Table 2 in [23] with a CD diagram using the Wilcoxon-Holm procedure implemented by [17]. We replaced the results for FCDD from [23] by our own and copied the others from the literature [10, 30, 25, 22, 21, 12, 35]. For each class, the methods are sorted by their respective ROC-AUC and assigned a ranking from 1 to 10 according to their position; then, every pair of methods is compared with the Wilcoxon signed rank test with the confidence level $\alpha = 5\%$. The CD diagram shows the average ranks of the methods on the horizontal scale, and the red bars group methods where each pair of methods are not significantly different according to the Wilcoxon signed rank test. The ranks from one to five (six to ten omitted for the sake of brevity) are shown in Table 4 in the Appendix A.

Figure 3 shows the test pixel-wise ROC-AUC scores during the optimization of the model used for MVTec-AD with the unsupervised setting. Due to time and resources constraints, we ran this experiment on 7 out of the 15 classes in MVTec-AD, each of them being evaluated 6 times (a few of them could not finish in time).

Experiments on MVTec-AD: unsupervised V.S. semi-supervised

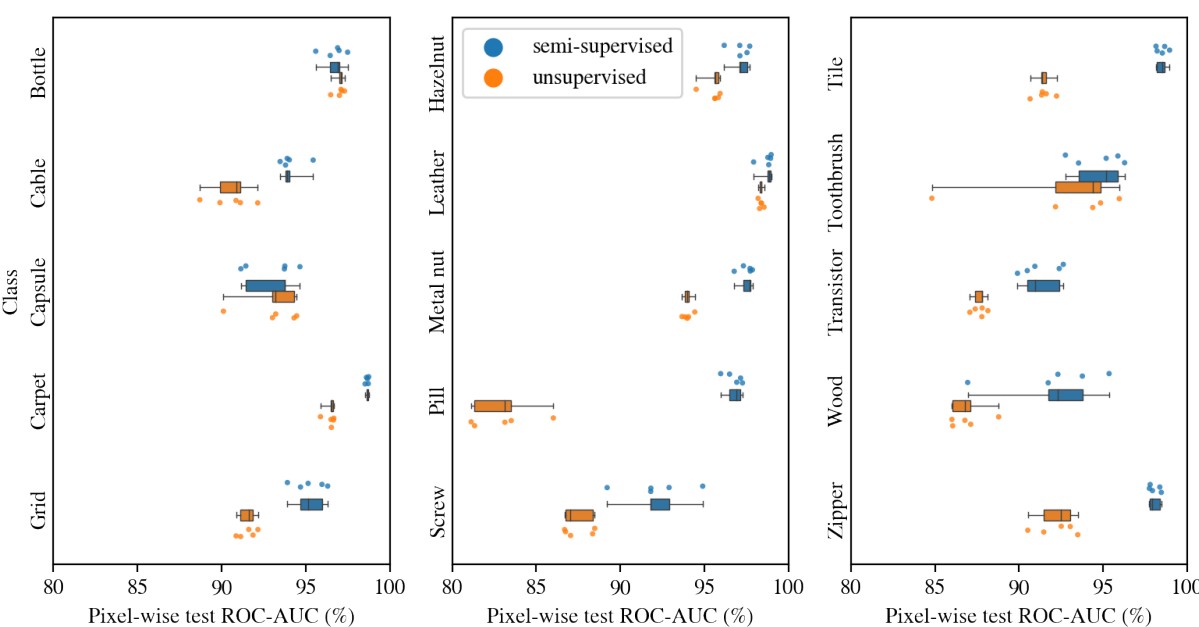

Figure 1: Our experiments on MVTec-AD: unsupervised and semi-supervised settings compared. We display a box plot of the performances (in terms of pixel-wise ROC-AUC on the test set) achieved in different runs along with their individual performances scattered on the $x$-axis.

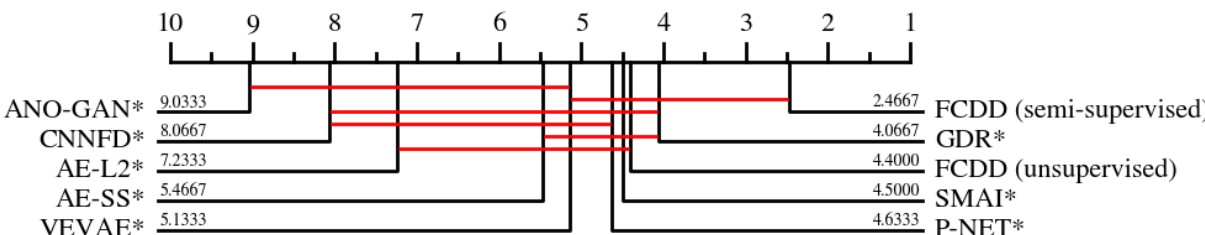

Figure 2: **MVTec-AD Critical Difference diagram**. Using the Table 2 from [23] with the results for FCDD replaced by our own, we build a critical difference diagram using the Wilcoxon-Holm method. Values on the scale are average rankings, and each red line groups a set of methods that are not significantly different in terms of ranking with a confidence level of $\alpha = 5\%$. The per-class ROC-AUC values used for FCDD are from our own experiments, and those marked with "*" were taken from the literature. References: Scores for Self-Similarity (AE-SS) [10], L2 Autoencoder (AE-L2) [10], AnoGAN [30], and CNN Feature Dictionaries (CNNFD) [25] were taken from Table 3 in [10]. Other scores were taken from their respective papers: Visually Explained Variational Autoencoder (VEVAE) from Table 2 in [22], Superpixel Masking and Inpainting (SMAI) from Table 2 in [21], Gradient Descent Reconstruction with VAEs (GDR) from Table 1 in [12], Encoding Structure-Texture Relation with P-Net for AD (P-NET) from Table 6 in [35].

## 5 Discussion

Our reproduction of the experiments closely agree with the quantitative results published in the original paper. The proposed setup is adapted to support the claims announced in the paper hold, and the results corroborate it. We obtained results consistently close to the published ones without any further tuning of the parameters or modification of the authors' code.

# MVTec-AD (unsupervised) training history

Pixel-level ROC-AUC on the test set measured during the training

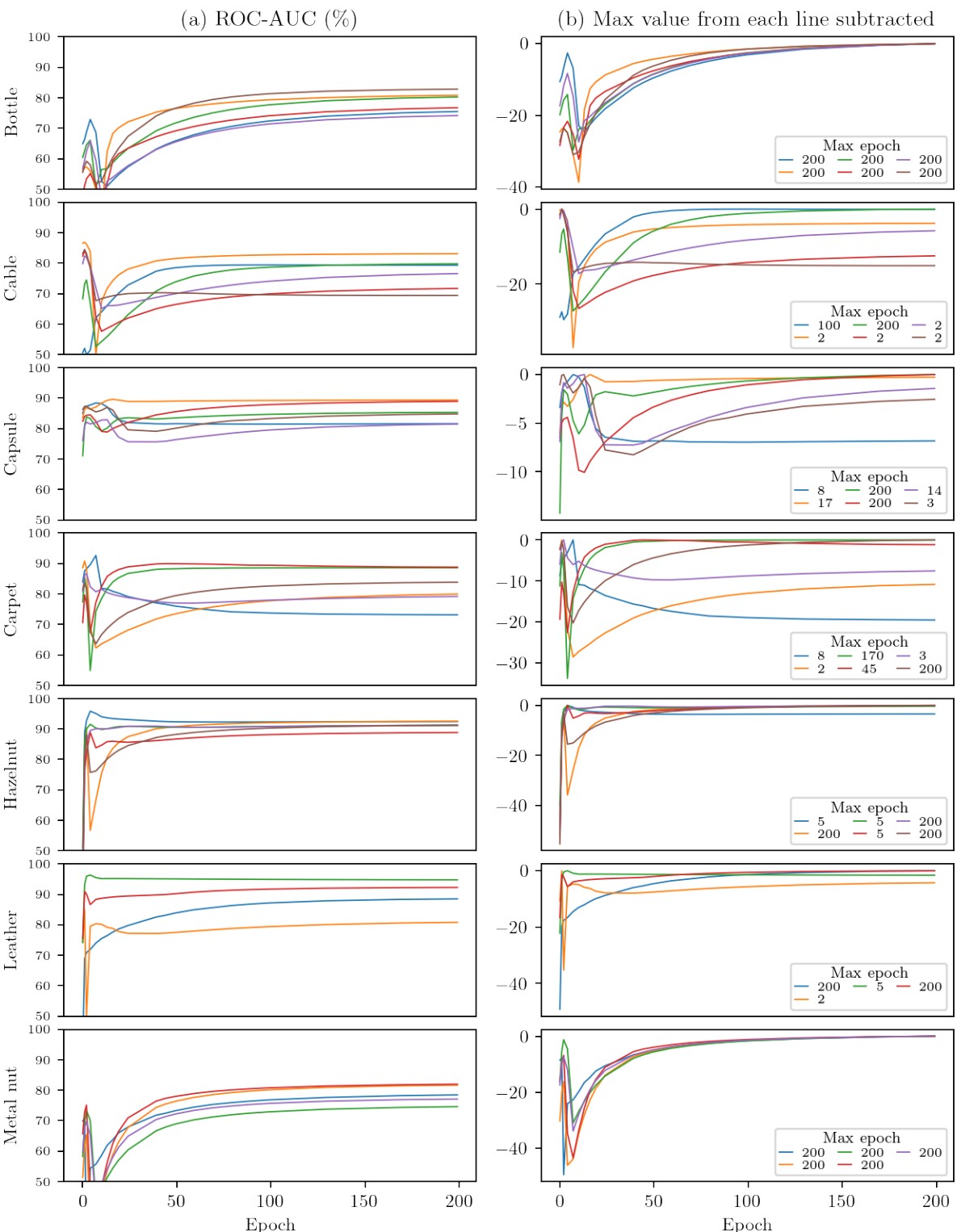

Performances were recorded at the following epochs (out of 1 to 200):
1, 2, 3, 5, 8, 11, 14, 17, 20, 25, 40, 45, 50, 60, 70, 80, 90, 100, 130, 170, 200

Figure 3: MVTec-AD test performance history.

## 5.1 What was easy

The paper is clear, and it was easy to grasp the core ideas presented in the main text. It also provided enough details about the experimental setup, including training hyper parameters and network architecture in the appendices.

The code was overall well organized and the instructions to use it were direct and easy to use. Conveniently, the experiments were well encapsulated scripts and default parameters matched those described in the text. In particular, the experiments are self-documenting (i.e. they keep record of the configurations, logs, and results, etc) and flexible, allowing the user to change (many) parameters without modifying the code.

## 5.2 What was difficult

**ImageNet** Using ImageNet was the hardest part. At first, it took about a month to get access to it on the official website. Then we had to find an alternative source [1] to find the correct version of ImageNet21k ("fall 2011") because it was not available on the official website anymore. Basic operations (e.g. decompressing data, moving files) proved challenging due to its size (1.2 TB compressed), and the instructions to manually prepare this dataset could be more explicit – we wasted several hours of work because of a few mistakes we made.

We could not run the experiments on that dataset with the same hyperparameters because the GPU we dispose of did not have enough memory (16GB). Although, we note that some solutions like using multiple GPUs or decreasing the batch size were possible but could not be tried in time.

**Minor code issues** There were a few minor bugs, which we corrected without considerable difficulty. They were mostly related with the script `add_exp_to_base.py`, which automatically configures and launches baseline experiments based on a previously executed one. Finally, the code structure was slightly overcomplicated; e.g. the levels of abstraction/indirection, specially heritage, could be simpler. Although, we stress that this negative point is minor and did not cause any critical issues.

## 5.3 Communication with original authors

We exchanged e-mails with the main author, mostly to ask for help with getting access to the right versions of ImageNet and executing the experiments on it. He replied promptly, his answers were certainly helpful, and we would like to express our sincere appreciation.

## 5.4 Beyond the paper

**MVTec-AD: supervision effect** The visualization proposed in Figure 1 further demonstrates that, with only a few images of real anomalies added to the training, the model's performance consistently improves. Only 4/15 classes have performance overlap, all others show a clear shift in the performance distribution.

However, it must be mentioned that the synthetic anomalies ignore the local supervision, therefore making its training sub-optimal. Figure 4 illustrates this with training images and their respective masks from the class "Pill": in 4a we see that the the semi-supervised setting provides pixel-level annotations on the anomalies (the ground truth mask), while in 4b we see that the entire image is considered anomalous in the unsupervised setting. This is a source of sub-optimality because, in the anomalous images, most pixels are, in fact, normal. In other words, similar images patches, free of synthetic anomaly, can be found both in normal and anomalous images.

Ultimately, this a clear opportunity of improvement that can bring the unsupervised setting's performance closer to the semi-supervised setting.

**Test performance history** Figure 3 reveals another issue with the method. Take, for instance, the purple and blue lines in the row "Carpet"; they reach a maximum point at the beginning of the gradient descent then converge to a point with less and less generalization power. These performance histories are evaluated on the test set, which is assumed to be unavailable at training time, so this information could not be used to stop the training or reject it. However, this reveals another opportunity of improvement because the training setting does not push the model to generalize well enough. Note, in Figure 4b, that the confetti noise also "stains" the background, creating synthetic anomalies "out of context", so using more realistic ones could be a solution?

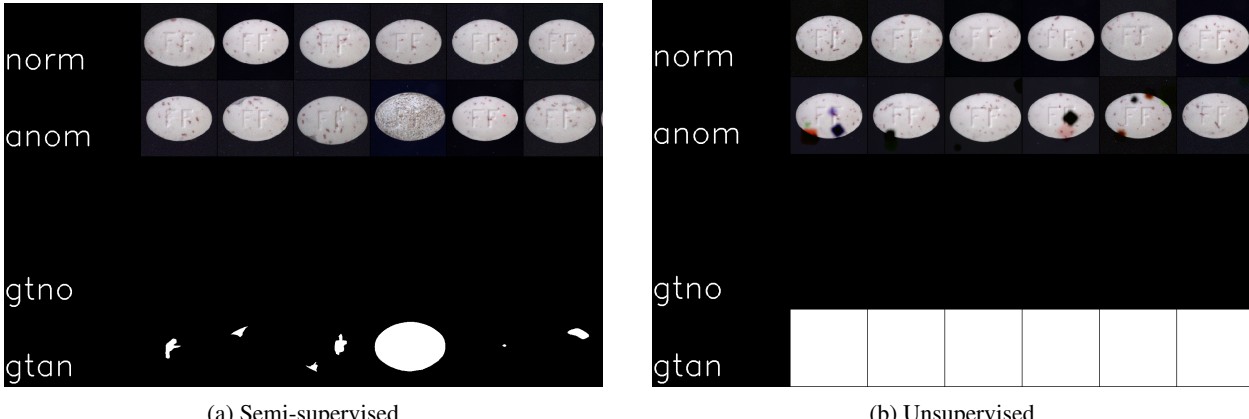

(a) Semi-supervised                (b) Unsupervised

Figure 4: MVTec-AD training images: unsupervised vs. semi-supervised.

We also see that some (good) performances are likely due to the pre-training (on ImageNet). For instance, the class "Hazelnut" often reaches its maximum test performance (or almost) with few epochs.

**Critical Difference (CD) diagram**     We propose a CD diagram in Figure 2 as a more appropriate methodology to aggregate results of all the classes. As a potential user is choosing an AD method for a new dataset, he or she is looking for the method(s) that will most likely be the best on his or her own problem. Therefore, the specific ROC-AUC scores on standard datasets have little importance but their relative performances is essential. In other words, what matters for a potential user is how the comparison of methods generalizes over specific datasets.

The experiments on MVTec-AD do not interact from one class (set as nominal) to another, making them essentially independent datasets; therefore, taking the average score over the classes may mislead the analysis. For instance, in [23], FCDD unsupervised is claimed to beat the state-of-the-art, although Figure 2 shows that GDR [12] has a better average ranking.

Note that the red bars in the diagram may give the impression that there is no relevant difference at all; although, it is important to observe that it was built considering only 15 datasets (therefore 15 rankings), making the statistical test hard, so using more datasets could refine these groups and provide better understanding. Finally, it is worth noting that the CD diagram is capable of incorporating new datasets, while the mean score over them would be too much affected if some cases are much easier or much harder than the others.

**State-of-the-art**     It is worth mentioning that more recent methods have claimed better results on the same benchmarks used in this work. For instance, at least 5 papers [27, 18, 34, 33, 15] claim to have a mean ROC-AUC above $98\%$ on the leaderboard for anomaly segmentation ("pixel-wise AD") on MVTec-AD in Papers with Code [8]. Unfortunately, we did not have the time to fully verify the experimental conditions in the sources but this serves as proxy evidence to take these results with consideration.

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

 **A Supplementary details**

| Normal Class | 5 | 4 | 3 | 2 | 1 |
|---|---|---|---|---|---|
| Bottle | GDR* | AE-SS* | **FCDD (SS)** | **FCDD (U)** | P-NET* |
| | 92.0 | 93.0 | 96.7 | 97.0 | 99.0 |
| Cable | VEVAE* | **FCDD (U)** | GDR* | SMAI* | **FCDD (SS)** |
| | 90.0 | 90.5 | 91.0 | 92.0 | 94.1 |
| Capsule | GDR* | **FCDD (SS)** | SMAI* | **FCDD (U)** | AE-SS* |
| | 92.0 | 92.9 | 93.0 | 93.0 | 94.0 |
| Carpet | VEVAE* | AE-SS* | SMAI* | **FCDD (U)** | **FCDD (SS)** |
| | 78.0 | 87.0 | 88.0 | 96.4 | 98.7 |
| Grid | AE-SS* | **FCDD (SS)** | GDR* | SMAI* | P-NET* |
| | 94.0 | 95.2 | 96.0 | 97.0 | 98.0 |
| Hazelnut | P-NET* | SMAI* | **FCDD (SS)** | GDR* | VEVAE* |
| | 97.0 | 97.0 | 97.1 | 98.0 | 98.0 |
| Leather | P-NET* | GDR* | VEVAE* | **FCDD (U)** | **FCDD (SS)** |
| | 89.0 | 93.0 | 95.0 | 98.4 | 98.7 |
| Metal nut | GDR* | SMAI* | **FCDD (U)** | VEVAE* | **FCDD (SS)** |
| | 91.0 | 92.0 | 94.0 | 94.0 | 97.5 |
| Pill | AE-SS* | P-NET* | SMAI* | GDR* | **FCDD (SS)** |
| | 91.0 | 91.0 | 92.0 | 93.0 | 96.8 |
| Screw | AE-L2* | AE-SS* | SMAI* | VEVAE* | P-NET* |
| | 96.0 | 96.0 | 96.0 | 97.0 | 100.0 |
| Tile | VEVAE* | **FCDD (U)** | CNNFD* | P-NET* | **FCDD (SS)** |
| | 80.0 | 91.4 | 93.0 | 97.0 | 98.5 |
| Toothbrush | VEVAE* | **FCDD (SS)** | SMAI* | GDR* | P-NET* |
| | 94.0 | 94.7 | 96.0 | 99.0 | 99.0 |
| Transistor | **FCDD (U)** | AE-SS* | **FCDD (SS)** | GDR* | VEVAE* |
| | 87.6 | 90.0 | 91.3 | 92.0 | 93.0 |
| Wood | GDR* | **FCDD (U)** | CNNFD* | **FCDD (SS)** | P-NET* |
| | 84.0 | 86.9 | 91.0 | 92.0 | 98.0 |
| Zipper | AE-SS* | P-NET* | SMAI* | **FCDD (U)** | **FCDD (SS)** |
| | 88.0 | 90.0 | 90.0 | 92.2 | 98.1 |

Table 4: Method rankings on MVTec-AD based on pixel-wise ROC-AUC. Using the Table 2 from [23], we compare the methods by normal class individually sorting the performances by pixel-wise ROC-AUC. The numbers in column names indicate the ranking (from 1 to 10); only the first 5 are displayed for the sake brevity. FCDD "unsupervised" and "semi-supervised" versions are respectively indicated by "(U)" and "(SS)", and their original values have been replaced by our own experiments' results.

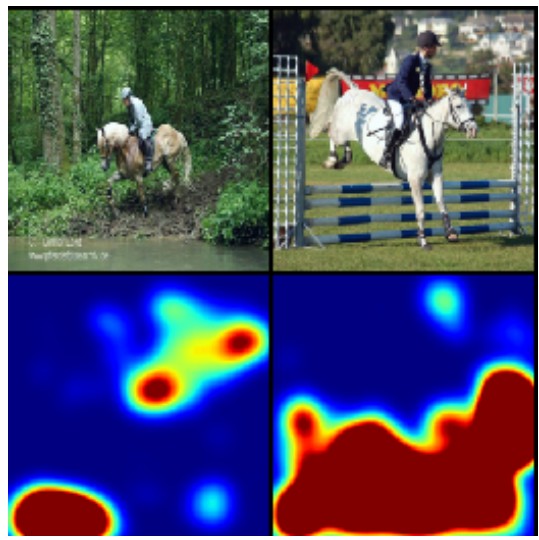

Figure 5: Heatmaps from the experiment on PASCAL-VOC where the Clever Hans effect can be observed.

