# OpenReview forum: "[Reproducibility Report] Explainable Deep One-Class Classification"
_ML_Reproducibility_Challenge/2021/Fall — Reject_

### Official Review · Reviewer_uZWz · 2022-03-01
**RE:[Reproducibility Report] Explainable Deep One-Class Classification**

**Rating:** 6
**Confidence:** 4

**Review:**

Summary:
This paper presents a report on the reproducibility of the fully Convolutional Data Description(FCDD) method for image anomaly detection(AD). This study verifies comparable performance of FCDD on fashion-MNIST and CIFAR-10 and outperforms existing methods on MVTec-AD on the pixel-wise task. However, additional analysis done in this paper suggests that the latter one should be claimed with consideration. Also, a qualitative study on PASCAL-VOC was peformed to confirm that FCDD can detect spurious model decisions by using built-in anomaly score heatmap.

Strengths:
•	For the experiments were done all performances almost matches with the ones reported in the original paper with a small marginal difference.
•	With visualizing comparison between unsupervised and semi-supervised performances on MVTec-AD, they verified that adding a few annotated images to the training results in improved performance.
•	Provided some opportunities for improvement.

Weaknesses:
•	Not reproducing results for the performance of ImageNet dataset.


Checkpoints:
-	Reproducibility Summary - Yes
-	Scope of reproducibility - Yes
-	Communication with original authors - Yes
-	Hyperparameter Search - No
-	Ablation Study - No
-	Discussion on results - Yes
-	Recommendations for reproducibility - Yes
-	Results beyond the paper - Yes
-	Overall organization and clarity - Yes

---

### Official Review · Reviewer_NN2e · 2022-03-01
**The paper conducts a thorough analysis of the original work. Although the authors do not use their own implementation of the original work, the paper still gives valuable insights into the approach.**

**Rating:** 7
**Confidence:** 4

**Review:**

The paper reproduces the results for an explainable anomaly detection approach for images. The results of the paper are that the authors were able to replicate per-class performances of the original paper.

The authors thoroughly explain the foundation of the paper as well as how they conducted each experiment. While the authors repeat unsupervised and semi-supervised experimental settings of the original paper, they also conduct additional analyses to better study potential drawbacks of the approach. Although the authors were not able to conduct experiments on ImageNet, the experiments are still very beneficial.

The authors report positive results with respect to being able to reproduce the original paper's claims. In addition, the authors report sub-optimal behaviour of the original approach when providing examples of real anomalies during training.

The only negative point of the proposed reproducibility paper is that the code for the approach was reused (with minor fixes) from the original paper. However, given the detailed experiments and analyses, the paper still adds sufficient value.

---

### Meta-Review · Program_Chairs · 2022-04-07

**Recommendation:** Reject
**Confidence:** 4

**Metareview:**

While the authors do reproduce the original results, they do so while reusing the original code and without any major changes to the approach. However, there are some missing elements, such as reproducing the results on ImageNet and a more in-depth discussion of the results obtained.

---

### Decision · Program_Chairs · 2022-04-09

Reject